# Symbolic Regression is NP-hard

**Marco Virgolin**                                                *marco.virgolin@cwi.nl*
*Centrum Wiskunde & Informatica, Amsterdam, the Netherlands*

**Solon P. Pissis**                                                *solon.pissis@cwi.nl*
*Centrum Wiskunde & Informatica, Amsterdam, the Netherlands*
*Vrije Universiteit, Amsterdam, the Netherlands*

**Reviewed on OpenReview:** *https://openreview.net/forum?id=LTiaPxqe2e*

## Abstract

Symbolic regression (SR) is the task of learning a model of data in the form of a mathematical expression. By their nature, SR models have the potential to be accurate and human-interpretable at the same time. Unfortunately, finding such models, i.e., performing SR, appears to be a computationally intensive task. Historically, SR has been tackled with heuristics such as greedy or genetic algorithms and, while some works have hinted at the possible hardness of SR, no proof has yet been given that SR is, in fact, NP-hard. This begs the question: Is there an exact polynomial-time algorithm to compute SR models? We provide evidence suggesting that the answer is probably negative by showing that SR is NP-hard.

## 1 Introduction

Symbolic regression (SR) is a sub-field of machine learning concerned with discovering a model of the given data in the form of a mathematical expression (or equation) (Koza, 1994; Schmidt & Lipson, 2009). For example, consider having measurements of planet masses $m_1$ and $m_2$, the distance $r$ between them, and the respective gravitational force $F$. Then, an SR algorithm would ideally re-discover the well-known expression (or an equivalent formulation thereof) $F = G \times \frac{m_1 m_2}{r^2}$, with $G = 6.6743 \times 10^{-11}$, by opportunely combining the mathematical operations (here, of multiplication and division) with the variables and constant at play.

The appeal of learning models as mathematical expressions goes beyond obtaining predictive power alone, as is commonplace in machine learning. In fact, SR models are particularly well suited for human interpretability and in-depth analysis (Otte, 2013; Virgolin et al., 2021b; La Cava et al., 2021). This aspect enables a safe and responsible use of machine learning models for high-stakes societal applications, as requested in the AI acts by the European Union and the United States (European Commission, 2021; 117th US Congress, 2022; Jobin et al., 2019). Moreover, it enables scientists to gain deeper knowledge about the phenomena that underlie the data. Consequently, SR enjoys wide applicability: SR has successfully been applied to astrophysics (Lemos et al., 2022), chemistry (Hernandez et al., 2019), control (Derner et al., 2020), economics (Verstyuk & Douglas, 2022), mechanical engineering (Kronberger et al., 2018), medicine (Virgolin et al., 2020b), space exploration (Märtens & Izzo, 2022), and more (Matsubara et al., 2022).

As we will describe in Sec. 2, many different algorithms have been proposed to address SR, ranging from genetic algorithms to deep learning ones. Existing algorithms either lack optimality guarantees or heavily restrict the space of SR models to consider. In fact, there is a wide belief in the community that SR is an NP-hard problem[1] (Lu et al., 2016; Petersen et al., 2019; Udrescu & Tegmark, 2020; Li et al., 2022). However, to the best of our knowledge, this belief had yet to be solidified in the form of a proof prior to the advent of this paper. Indeed, we prove that there exist instances of the SR problem for which one cannot discover the best-possible mathematical expression in polynomial time unless P=NP. Id est, SR is an NP-hard problem.

---

[1]Lu et al. (2016) state that SR is NP-hard but provide no reference nor proof.

## 2 Background

We begin with a historical overview of how SR has been attempted from an algorithmic perspective (Sec. 2.1), and then follow with related work concerning hardness (Sec. 2.2).

### 2.1 SR algorithms

The introduction of SR is generally attributed to John R. Koza (e.g., Zelinka et al. (2005) make this claim); however, the problem of finding a mathematical expression or equation that explains empirical measurements was already considered in earlier works (Gerwin, 1974; Langley, 1981; Falkenhainer & Michalski, 1986). Such works build mathematical expressions by iterative application of multiple heuristic tests on the data.

Koza is best known for his pioneering work on genetic programming (GP), i.e., the form of evolutionary computation where candidate solutions are variable-sized and represent programs (Koza et al., 1989; Koza, 1990; 1994). Early forms of GP were proposed by Cramer (1985); Hicklin (1986). Koza showed that GP can be used to discover SR models by encoding mathematical expressions as computational trees (see Fig. 1). In such trees, internal nodes represent functions (e.g., $+$, $-$, $\times$, etc.) that are drawn from a pre-decided set of possibilities, and leaf nodes represent variables or constants (e.g., $x_1$, $x_2$, ..., $-1$, $\pi$, etc.). GP evolves a population of trees by initially sampling random trees, and then conducts the following steps: (1) stochastic replacement and recombination of their sub-trees; (2) evaluation of the fitness by executing the trees and assessing their output; and (3) stochastic survival of the fittest.

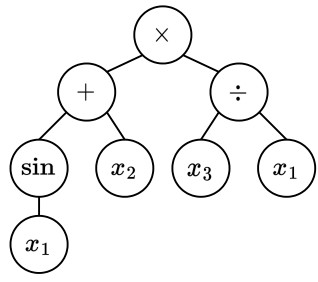

Figure 1: Example of a tree that encodes $f(\mathbf{x}) = (\sin(x_1) + x_2) \times x_3/x_1$.

Recently, La Cava et al. (2021) proposed *SRBench*, a benchmarking platform for SR that includes more than 20 algorithms and more than 250 data sets. SRBench shows that several state-of-the-art algorithms for SR are GP-based. Among these, at the time of writing, *Operon* by Burlacu et al. (2020) was found to perform best in terms of discovering accurate SR models; and *GP-GOMEA* by Virgolin et al. (2021a) was found to perform best in terms of discovering decently-accurate and relatively-simple SR models (i.e., shorter mathematical expressions). Other forms of GP, such as *strongly-typed* GP (Montana, 1995), *grammar-guided* GP (McKay et al., 2010), and *grammatical evolution* (O'Neill & Ryan, 2001), are often used to tackle *dimensionally-aware* SR, i.e., the search of mathematical expressions with constraints to achieve meaningful combinations of units of measurement.

SR has been addressed with other types of algorithms than genetic ones, including, e.g., Monte-Carlo tree search (Cazenave, 2013; Sun et al., 2022). Moreover, several authors proposed deterministic algorithms. For example, Worm & Chiu (2013) and Kammerer et al. (2020) proposed enumeration algorithms which make SR tractable by restricting the space of possible models to consider and including dynamic programming and pruning strategies. Cozad (2014); Cozad & Sahinidis (2018) showed how SR can be addressed with mixed integer nonlinear programming. McConaghy (2011) proposed *FFX*, which generates a linear combination of many functions that are linearly-independent from each other, and then fits its coefficients with the *elastic net* (Zou & Hastie, 2005) to promote sparsity. Olivetti de França (2018) and Rivero et al. (2022) propose greedy algorithms that start from small mathematical expressions and iteratively expand them, by replacing existing components with larger ones from a set of possibilities.

Lastly, recent years have seen the proposal of deep learning-based algorithms for SR. Petersen et al. (2020) cast the SR problem as a reinforcement learning one and train a recurrent neural network to generate accurate SR models. Udrescu & Tegmark (2020) leverage neural networks in order to test for symmetries and invariances in the data that are then used to prune the space of possible SR models. An end-to-end approach is taken by Kamienny et al. (2022) and Vastl et al. (2022), who train deep neural transformers to produce SR models directly from the data. Li et al. (2022) seek SR models by proposing a convexified formulation of deep reinforcement learning.

In summary, existing SR algorithms are either heuristics, which do not guarantee optimality (e.g., genetic, greedy, or deep learning-based algorithms), or they are exact algorithms that achieve optimality but only over

a small subset of all possible SR models, to bound the runtime (e.g., dynamic programming and mixed-integer nonlinear programming algorithms). This strongly hints to the fact that SR is NP-hard. As mentioned earlier, no proof has yet been given.

## 2.2 Related hardness results

SR is typically posed as an empirical risk minimization (ERM) problem or, when regularization is considered, a structural risk minimization one (Vapnik, 1999). There exists a multitude of theoretical results in machine learning posing the problem as an ERM one. For example, Blum & Rivest (1992) famously proved the NP-completeness of training a three node-neural network to label a given data set correctly. The loss employed in such situations is commonly the 0-1 loss, i.e., the loss that returns 0 if the output of the model (or *prediction*) equals the output that is expected from the data (or *label*) and 1 otherwise. More recently, it has been shown that, under the 0-1 loss, it is NP-hard to even train a linear classifier to be $\epsilon$-better than random (Feldman et al., 2012). In the context of coding theory, ERM with 0-1 loss has been used to prove the NP-hardness of finding a univariate polynomial of maximum degree $k$ over a finite field (a *code*) that maximizes code matching (Guruswami & Vardy, 2005).

Under different types of loss, polynomial-time solutions to ERM exist. A famous example of this is linear regression under the squared error loss, which can be solved in polynomial time via ordinary least squares. Recently, Backurs et al. (2017) presented fine-grained complexity results for ERM with kernel-based and neural network-based approaches.

SR can be set to search in the space of polynomials and one can choose to use the 0-1 loss. Moreover, one can in principle set SR to work in finite fields rather than on real numbers to operate with polynomials for discrete codes. For example, Koza (1990) shows how to set GP to learn Boolean circuits by composition of logic gates. This means that the hardness of SR can follow from linking back to results such as the one by Feldman et al. (2012) (we sketch how this can be achieved at the end of Sec. 4). The modern connotation of SR is focused on regression (i.e., we seek a model $f : \mathbb{R}^d \to \mathbb{R}$ with $d$ the number of features) and commonly-used losses have co-domain in $\mathbb{R}_0^+$, such as the absolute error loss or the squared error loss; rather than the 0-1 loss. We will provide a proof of NP-hardness that is general to this type of losses (Eq. (2)). In essence, we will show that when certain *basic* arithmetic operations are chosen for combining (e.g., addition), SR becomes NP-hard. This choice of basic operations allows us to reduce from a rather classical variant of the knapsack problem, namely, the *unbounded subset set* problem (Kellerer et al., 2004).

## 3 Preliminaries

We will hereon refer to SR models as *functions* when appropriate, as this is their fundamental nature. Functions take variables as arguments. One can use the *identity function*, i.e., the function that returns the value of the variable taken as its argument. For simplicity of exposition, we will generically refer to functions and not make a distinction between (non-identity) functions and variables. Similarly, we will refer to functions also for *constant functions*, i.e., functions that can only return a single numerical value, irrespective of their arguments. This said, let us recall the concept of function composition, which is central to SR.

**Definition 1.** Function composition

*Given two functions $f : \mathbb{A} \to \mathbb{B}$ and $g : \mathbb{B} \to \mathbb{C}$, function composition, which we denote by $g \circ f$, is the operation that produces a third function $h : \mathbb{A} \to \mathbb{C}$, such that $h(x) = g(f(x))$.*

Thanks to function composition, we can now define the concept of *search space* of an SR problem.

**Definition 2.** Search space of SR

*Let $\mathcal{P}$ be a set of functions. The search space of SR is the function space $\mathcal{F}$ that contains all functions that can be formed by composition of the elements of $\mathcal{P}$.*

To better understand what Def. 2 states, consider that $\mathcal{P}$ can be set to contain a mix of functions that perform basic algebraic operations such as addition, subtraction, multiplication, and division; transcendental functions such as sin, cos, log, exp; constant functions (or simply constants), such as $c_{42}(x) = 42$ and $c_\pi(x) = \pi$ for

any $x$; and identity functions that represent variables of interest for the problem at hand, such as $x_1, x_2, x_3$. $\mathcal{P}$ is typically referred to as the *primitive set*, and its elements as *primitives* (Poli et al., 2008). Once $\mathcal{P}$ has been decided, $\mathcal{F}$ is determined. For example, choosing $\mathcal{P} = \{+(\cdot, \cdot), -(\cdot, \cdot), \times(\cdot, \cdot), x_1, x_2, -1, +1\}$ means that $\mathcal{F}$ will contain a subset of all possible polynomials of arbitrary degree in $x_1$ and $x_2$. In particular, $\mathcal{F}$ is a subset because only some coefficients can be expressed, by composing constants with addition, subtraction, and multiplication.

Let us clarify a point regarding constants in particular. Normally, one would include constants which are relevant to the instance of SR at hand. For example, if the unknown phenomenon for which an SR model is sought is suspected to have sinusoidal components, it may be advisable to include multiples of $\pi$ in $\mathcal{P}$. Moreover, $\mathcal{P}$ can be set to contain special elements that represent probability distributions from which constants can be sampled (see the concept of *ephemeral random constant* described by Koza (1994); Poli et al. (2008)). We denote one such element by $\mathfrak{R}$ and, e.g., $\mathfrak{R}$ can be chosen to represent the uniform distribution between two numbers, or the normal distribution with a certain mean and variance. When an SR algorithm picks $\mathfrak{R}$ from $\mathcal{P}$ to compose an SR model, a constant is sampled from the distribution identified by $\mathfrak{R}$. Here (more specifically, in Corollary 1) we will generously assume that any constant can be sampled directly from $\mathfrak{R}$, and therefore that there is no need for a real-valued optimizer to be part of the SR algorithm. For example, having $\mathcal{P} = \{+(\cdot, \cdot), -(\cdot, \cdot), \times(\cdot, \cdot), x_1, x_2, \mathfrak{R}\}$ will mean that $\mathcal{F}$ contains *all* polynomials of arbitrary degree in $x_1$ and $x_2$.

We can now proceed by providing a definition of the SR problem. While this definition can be extended to other domains, we focus on handling real-valued numbers as the majority of the works takes place in this domain, and subsets thereof.

**Definition 3.** Symbolic Regression (SR) problem

*Given a set $\mathcal{P}$ of functions, a distance $\mathcal{L} : \mathbb{R} \times \mathbb{R} \to \mathbb{R}_0^+$, vectors $\mathbf{x}_i = (x_{1,i}, \ldots, x_{d,i}) \in \mathbb{R}^d$ and scalars $y_i \in \mathbb{R}$, for $i = 1, \ldots, n$, the SR problem asks for finding a function $f^\star$ such that:*

$$f^\star \in \operatorname*{arg\,min}_{f \in \mathcal{F}} \frac{1}{n} \sum_{i=1}^{n} \mathcal{L}\left(y_i, f(\mathbf{x}_i)\right) \tag{1}$$

*where $\mathcal{F}$ is the search space that is defined by $\mathcal{P}$.*

We provide some remarks concerning the proposed definition of the SR problem. Firstly, let us map the objects provided in the definition to terms familiar to a machine learning audience. The pair $(\mathbf{x}_i, y_i)$ is normally what is referred to as *observation*, *data point*, *example*, or *sample*, where $x_{j,i}$ is the value of the $j$th *feature* or *variable* for the $i$th observation, and $y_i$ is the value of the *label* or *target variable* for the same observation. The set that contains the observations upon which $\mathcal{L}$ is computed, i.e., $\mathcal{D} = \{(\mathbf{x}_i, y_i)\}_{i=1}^{n}$, is called *training set*. It is commonly assumed that the observations in $\mathcal{D}$ were drawn independently and identically distributed (i.i.d.) from an unknown probability distribution. Moreover, the distance $\mathcal{L}$ is called *loss*. Losses need not be distances, but in SR they normally are. Popular choices in the literature are the absolute error loss and the squared error loss. Here, we consider losses of the form:

$$\mathcal{L}(y_i, f(\mathbf{x}_i)) = |y_i - f(\mathbf{x}_i)|^r, \tag{2}$$

with $r \in \mathbb{N}_0$: for $r = 0$ one gets the 0-1 loss; for $r = 1$ one gets the absolute error loss; and for $r = 2$ one gets the squared error loss.

The minimization of the loss function across the observations in $\mathcal{D}$ makes SR an ERM problem. As is generally the case for learning, one actually desires $f$ to *generalize* to *new* (or also called *unseen*) observations, i.e., observations that come from the same underlying probability distribution but were not available in $\mathcal{D}$. In other words, it is not sufficient that $f^\star$ is a best-possible function with respect to the training set, as the loss should remain minimal also for new observations that are not available to us. To this end, a common practice is to heuristically use a separate set of data (the *validation set*) to estimate the generalization to observations outside the training set. Another approach, which is often used together and not alternative to adopting a validation set, is to perform *structural risk minimization*, i.e., account for regularization terms such as

$\lambda \times C(f)$, where $\lambda \in \mathbb{R}_0^+$ controls the regularization strength and $C : \mathcal{F} \to \mathbb{R}$ is a function of the complexity of $f$. Typical goals of such regularization terms are improving generalization (by limiting effects akin to Runge's phenomenon (Fornberg & Zuev, 2007)) or, particularly for SR, improving the interpretability of $f$. Implementations of $C$ range from weighted counting of the number of primitives that constitute $f$ (Ekart & Nemeth, 2001; Hein et al., 2018) to machine learning models trained from human feedback to predict $f$'s interpretability (Virgolin et al., 2020a; 2021b). Here, for simplicity, we do not consider regularization and focus on ERM alone. Equivalently put, we consider $\lambda = 0$: note that this choice is not limiting because this step can also be taken when constructing the proof of Theorem 1. We will briefly get back to how $\lambda > 0$ and $C$ may be used to build interesting search spaces for the hardness of SR at the end of Sec. 4.

Still, considering a "pure optimization" formulation (or ERM), as given in Eq. (1), can be considered to be a pre-requisite for being able to machine-learn accurate models from the data; in fact, it is commonplace for literature that concerns the hardness of learning to provide results with respect to the training set (see, e.g., (Feldman et al., 2012; Hu et al., 2019)). In a similar fashion, here we will consider the case of minimizing the empirical risk with respect to the training set $\mathcal{D}$ and show that this alone already poses a challenge for any SR algorithm.

Here, we assume that computing $f(\mathbf{x})$ and $\mathcal{L}(\mathbf{y}, f(\mathbf{x}))$ (see Def. 3) can be done in polynomial time. Regarding $\mathcal{L}$, our assumption is met for commonly-used losses such as the absolute and squared error ones. In fact, computing losses of such form takes $O(n)$ operations, i.e., the runtime is linear in the number of observations. Regarding $f$, our assumption is met, e.g., for all functions that can be discovered by the SR algorithms currently in SRBench (La Cava et al., 2021); a notable exception of practical interest are recursive functions taking exponential time to compute (see, e.g., d'Ascoli et al. (2022)). For non-recursive functions, $f$ can be implemented as a directed acyclic graph, where nodes represent the functions from $\mathcal{P}$, and edges represent compositions. To compute $f(\mathbf{x})$, it suffices to visit each node of the graph for each observation, thus requiring $O(\ell \times n)$ operations, where $\ell$ is the number of primitives in $f$. Fig. 1 shows an example of such a graph, especially in the form of a *tree*, which is perhaps the most common way of encoding mathematical expressions in SR (see, e.g., the SR algorithms benchmarked by La Cava et al. (2021)).

We conclude this section with the following important definition.

**Definition 4.** Decision version of the SR problem (SR-Dec)

*Given an SR instance and an $\epsilon \in \mathbb{R}_0^+$, SR-Dec outputs* YES *if and only if:*

$$\exists f \in \mathcal{F} : \frac{1}{n} \sum_{i=1}^{n} \mathcal{L}\left(y_i, f(\mathbf{x}_i)\right) < \epsilon. \tag{3}$$

Essentially, Def. 4 is the problem of deciding whether there exists a function $f$ in the search space such that its empirical risk is smaller than a chosen threshold $\epsilon$.

## 4 The result

We proceed directly by providing the main result of this paper.

**Theorem 1.** *The SR problem is NP-hard.*

*Proof.* Let us begin by stating that SR-Dec is in NP. Recall that the computations of $f(\mathbf{x})$ and $\mathcal{L}(y_i, f(\mathbf{x}_i))$ take polynomial time (see Sec. 3). Of course, the check $< \epsilon$ takes $O(1)$ time. Thus, if $f$ is guessed by an oracle, then we can provide an answer to SR-Dec in polynomial time.

We proceed by considering the unbounded subset sum problem (USSP). USSP is a similar problem to the unbounded knapsack problem, where a same item can be put in the knapsack an arbitrary number of times, and the weight of an item corresponds exactly to the profit gained by including that item in the knapsack. The decision version of USSP, USSP-Dec, is defined as follows. Given $j = 1, \ldots, k$ ($k$ items), $w_j \in \mathbb{N}$ (weight

of that item), and $t \in \mathbb{N}$ (the target), USSP-Dec asks:

$$\exists \mathbf{m} : \sum_{j=1}^{k} w_j m_j = t? \tag{4}$$

where $m_j \in \mathbb{N}_0$ (multiplicity with which an item is picked). USSP-Dec is known to be NP-complete (Kellerer et al., 2004).

To prove that SR-Dec is NP-complete, we show that any instance of USSP-Dec can be reduced to some instance of SR-Dec in polynomial time. To this end, we will restrict SR-Dec as follows: (1) We pick the set of primitives $\mathcal{P}$ to be $\mathcal{P} = \{+, x_1, \ldots, x_d\}$, with $d = k$; (2) We set $\epsilon = 1$. Next, we craft $\mathcal{D}$ to have a single observation ($n = 1$) and $d = k$ features. For the only observation in $\mathcal{D}$ (dropping the index for the observation number, since there is only one), we set $x_1 = w_1, x_2 = w_2, \ldots, x_k = w_k$, and $y = t$.

In other words, we have set the search space $\mathcal{F}$ to contain only linear sums of the features in the data set $\mathcal{D}$, i.e., functions of the form $f(\mathbf{x}) = \sum_{j=1}^{d} x_j m_j$. Importantly, $m_j \in \mathbb{N}_0$ and $x_i \in \mathbb{N}$, meaning that the co-domain of any $f$ is $\mathbb{N}_0$. Consequently, the smallest non-zero loss that can be achieved is 1. The only functions $f$ that can achieve an error smaller than $\epsilon = 1$ are those that interpolate the observation exactly, i.e., $f(\mathbf{x}) = y$.

Then, the following holds:

$$(Eq.~(3)~with~\epsilon = 1) \quad \exists f \in \mathcal{F} : \mathcal{L}\left(y, f(\mathbf{x})\right) < 1? \tag{5}$$

$$(\mathcal{L}\left(y, f(\mathbf{x})\right) < 1 \iff f(\mathbf{x}) = y) \quad \exists f \in \mathcal{F} : f(\mathbf{x}) = y? \tag{6}$$

$$(Equivalence~y = t~due~to~\mathcal{D}) \quad \exists f \in \mathcal{F} : f(\mathbf{x}) = t? \tag{7}$$

$$(Expanding~\mathcal{F}~based~on~choice~of~\mathcal{P}) \quad \exists f \in \left\{ \sum_{j=1}^{d} x_j m_j : m_j \in \mathbb{N}_0 \right\} : f(\mathbf{x}) = t? \tag{8}$$

$$(Equivalence~x_j = w_j, d = k~due~to~\mathcal{D}) \quad \exists f \in \left\{ \sum_{j=1}^{k} w_j m_j : m_j \in \mathbb{N}_0 \right\} : f(\mathbf{x}) = t? \tag{9}$$

$$(Re\text{-}formulating~in~terms~of~\mathbf{m}) \quad \exists \mathbf{m} : \sum_{j=1}^{k} w_j m_j = t? \tag{10}$$

In other words, there exist some instances of SR-Dec that can be re-formulated as USSP-Dec (cfr. Eqs. (4) and (10)). Now, since assembling $\mathcal{P}$ as stated above takes linear time in $k$, picking $\epsilon = 1$ takes $\mathcal{O}(1)$ time, and constructing $\mathcal{D}$ as stated above takes linear time in $k$, then any instance of USSB-Dec can be reduced to some instance of SR-Dec in polynomial time: SR-Dec is NP-complete.

We conclude the proof with a *reductio ab absurdum*. Let us assume that there exists an algorithm to compute an optimal $f^\star$ for the SR problem (Def. 3) in polynomial time. An optimal $f^\star$ is the one for which the loss is minimal, which means that using $f^\star$ in Eq. (3) allows us to immediately answer SR-Dec. Since verifying that $\mathcal{L}(\mathbf{y}, f^\star(\mathbf{x})) < \epsilon$ takes polynomial time, we conclude that if the SR problem can be solved in polynomial time, then we can also solve SR-Dec in polynomial time. Therefore, the SR problem is NP-hard.

$\square$

We remark that, in the proof of Theorem 1, we construct $\mathcal{P}$ so as not to contain $\mathfrak{R}$ (nor any constant). Some readers might disagree with this quite broad definition of SR. In fact, some SR algorithms heavily rely on the presence of constants as well as on their optimization (e.g., *FFX* by McConaghy (2011) and *FEAT* by La Cava et al. (2018)). Not allowing for arbitrary constants to be present in the functions of the search space might be seen as a violation of the very definition of SR. In other words, some might think that $\mathcal{P}$ *must* contain $\mathfrak{R}$. We next show that SR remains NP-hard in this special case.

**Corollary 1.** *The SR problem is NP-hard even when $\mathcal{P}$ must include $\mathfrak{R}$.*

*Proof.* We follow a similar construction of the proof of Theorem 1. This time, we set $\mathcal{P}$ to additionally contain $\mathfrak{R}$, i.e., $\mathcal{P} = \{+, x_1, x_2, \ldots, x_d, \mathfrak{R}\}$, with $d = k$. This means that the function space $\mathcal{F}$ now contains functions of the form $f(\mathbf{x}) = c + \sum_{j=1}^d x_j m_j$ with $m_j \in \mathbb{N}_0$ and $c \in \mathbb{R}$ (sampled from $\mathfrak{R}$). As to $\mathcal{D}$, we will now include two observations instead of a single one. The first observation is set as before, i.e., $x_{1,1} = w_1, x_{2,1} = w_2, \ldots, x_{k,1} = w_k$ ($d = k$) and $y_1 = t$. For the second observation, we set $x_{1,2} = 0, x_{2,2} = 0, \ldots, x_{k,2} = 0$ and $y_2 = 0$, i.e., the value of all features and of the label are set to zero. It now remains to determine how we should set $\epsilon$.

Because of our construction of $\mathcal{D}$, an $f$ for which SR-Dec answered YES in the situation considered in Theorem 1, i.e., with $c = 0$, would still make SR-Dec answers YES for a $c$ with sufficiently small magnitude. Note that those functions interpolate $\mathcal{D}$ exactly if $c = 0$. Thus, the magnitude of $c$ must be such that $|c|^r < \epsilon$, with $r$ the degree of the loss (Eq. (2)), because the corresponding empirical risk for those functions is $\frac{1}{2}|c|^r + \frac{1}{2}|c|^r = |c|^r$, where the two summands on the left side of the equation are respective to the two observations in $\mathcal{D}$.

Now the question becomes whether using $c \neq 0$ allows to answer YES to more functions than those that would interpolate $\mathcal{D}$ when $c = 0$. If that would be true, then we can no longer apply the strategy used in the proof of Theorem 1 to reduce from USSP-Dec. To be able to still use that strategy, we will now show that there exist instances of SR-Dec, in particular by picking a different $\epsilon$, such that even if $\mathcal{P}$ *must* include $\mathfrak{R}$, then only functions with $c = 0$ are candidates for a YES answer. This would allow us to reduce once again from USSP-Dec (by appropriately picking $\epsilon$), because $f(\mathbf{x}) = 0 + \sum_{j=1}^d x_j m_j = \sum_{j=1}^d x_j m_j$, as in Eq. (8).

For $c \neq 0$ to allow SR-Dec to answer YES to more functions than those for when $c = 0$, $c$ must contribute to lower the empirical risk.

For the second observation, $c$ can only increase the risk, because the loss is:

$$\mathcal{L}(y_2, f(\mathbf{x}_2)) = |y_2 - f(\mathbf{x}_2)|^r = \left| 0 - \left( c + \sum_{j=1}^d 0 \times m_j \right) \right|^r = |c|^r. \tag{11}$$

For the first observation, however, using $c \neq 0$ can lower the respective loss and thus contribute to lowering the empirical error. In particular, consider that the first functions that are candidates to receive a YES answer thanks to $c \neq 0$ are those that had a loss of 1 when $c = 0$ (see the proof of Theorem 1, where we set $\epsilon = 1$). If we can have SR-Dec answer NO to these functions, then it will necessarily answer NO also to all other functions that have a loss larger than 1 on the first observation when $c = 0$. We thus proceed by considering that the smallest, non-zero loss that can be obtained on the first observation for $c \neq 0$, is $|1 - c|^r$. This leads to the following empirical risk over our $\mathcal{D}$:

$$\frac{1}{2}\sum_{i=1}^2 \mathcal{L}(y_i, f(\mathbf{x}_i)) = \frac{1}{2}\left(|1 - c|^r + |0 - c|^r\right). \tag{12}$$

For any integer $r \geq 0$, the minimum is $\frac{1}{2} \times \frac{1}{2^{r-1}} = 2^{-r}$: this is easy to verify for $r = 0$ and $r = 1$, while for $r \geq 2$ Eq. (12) describes a U-shaped curve that is symmetric around, and has minimum in $c = \frac{1}{2}$. Therefore, it suffices to pick $\epsilon = 2^{-r}$: even if $c$ is optimal, one cannot lower the empirical risk below $2^{-r}$ for any function whose loss (on the first observation) is not zero. In other words, imposing $\epsilon = 2^{-r}$ ensures that SR-Dec will answer YES *iff* SR-Dec answers YES also for $c = 0$. This means that one can set $c = 0$ and proceed with a reduction from USSP-Dec as in the proof of Theorem 1. $\square$

Finally, we provide the following remark.

**Remark.** *One can consider the structural risk minimization setting whereby the following minimization is sought:*

$$\frac{1}{n}\sum_{i=1}^n \mathcal{L}(y_i, f(\mathbf{x}_i)) + \lambda C(f), \ \ with \ \lambda > 0. \tag{13}$$

*Then, SR-Dec can be restricted to automatically answer NO for any $f$ that does not satisfy certain conditions, such as linearity. For example, one can pick $\mathcal{P} = \{+, \times, x_1, \ldots, x_d, \mathfrak{R}\}$ to search in the space of arbitrary*

*polynomials, and pick $C$ such that $C(f) = \infty$ if $\deg f \geq k$ else $0$, for an arbitrary integer $k \geq 0$. Combining this with the use of the 0-1 loss function enables to reduce SR-Dec from existing results in literature that consider linear classifiers (Feldman et al., 2012) or coding polynomials in finite fields (Guruswami & Vardy, 2005).*

## 5 Conclusion

Our main contribution here was to prove that symbolic regression (SR), i.e., the problem of discovering an accurate model of data in the form of a mathematical expression, is in fact NP-hard. In particular, we have provided formal definitions of what SR entails, and showed how the decision version of the unbounded subset sum problem can be reduced to a decision version of the SR problem. Except for the general definition of SR we considered, we have additionally shown that SR remains NP-hard even when the set of primitives must contain distributions from which constants can be sampled, and provided a sketch of how an alternative proof can be constructed by using the 0-1 loss and previous results from the literature.

Having settled the matter on the hardness of SR, we hope that this note inspires further works on lower and upper bounds of different SR variants. In fact, while we have shown that hardness holds in principle (by picking a search space suitable for reduction from USSP-Dec), there might exist specific variants of SR (e.g., different search spaces, specific regularization terms, or specific type of data) that are more commonly encountered in practical applications. For such more specific variants, proving hardness or designing polynomial-time algorithms would complement the SR status quo, which mostly focuses on heuristic algorithmic design. Ultimately, theoretical advances may greatly help advancing our knowledge of what is possible with SR.

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
