# OpenReview forum: "Symbolic Regression is NP-hard"
_TMLR — Accepted by TMLR_

### Review · Reviewer_YHbt · 2022-08-02

**Summary Of Contributions:**

Symbolic regression is one of the central techniques for interpretable and verifiable ML in many applications. So far, using evolutionary algorithms (EA) is one of the most successful approaches. However, with the black-box view of EAs, there comes the question of whether a polynomial algorithm could exist to find an optimal solution. The authors of the paper at hand show that by reduction of the unbounded subset sum problem, the symbolic regression problem is NP-hard and thus, it is unlikely that an optimal solution can be found efficiently.

**Requested Changes:**

A bit nitpicking on the wording: "this is already problematic for any SR algorithms". I wouldn't say that this is problematic because SR algorithms are nevertheless fairly successful. Maybe let's say that it already poses a challenge to do this on the training set efficiently.

Is it guaranteed that there is a unique solution to Equation 1? If not, I wonder whether it should be f^* \in \argmin since the task would not to find all solutions, but only one of them.


**Strengths And Weaknesses:**

### Strengths

* It is an important result in showing that SR is NP-hard
* The paper is very well written and easy to follow
* The reduction proof seems to be correct

### Weaknesses

* Two simplifications were assumed (i.e., no regularization and no recursive functions). However, I agree with the authors that this does not diminish the importance of their result.

---

> ### Author Response · Authors · 2022-09-06
> **Response to Reviewer YHbt**
>
> We are thankful for the reviewer's suggestions.
>
> We would like to address the mentioned Weaknesses, then proceed with addressing the requested changes.
>
> ### Weaknesses
> > Two simplifications were assumed (i.e., no regularization and no recursive functions). However, I agree with the authors that this does not diminish the importance of their result.
>
> Thank you for agreeing that these aspects do not diminish the importance of our result.
> - Regarding recursive functions, we tried to improve our explanation regarding the fact that we do not consider recursive functions (some of which can take exponential runtime).
> - Regarding regularization, we took two actions.
>   1. We try to convey better in the paper that, in our opinion, choosing not to consider regularization is not really a limiting factor, because one can pick $\lambda=0$ during the construction of the proof to proceed with reducing from USSP-Dec. We now write:
>   > “Note that this choice is not limiting because this step can also be taken when constructing the proof of Theorem 1”.
>   2. Also, we extended the end of Section 4 with a sketch of how one we make active use of regularization ($C$) to limit the types of function to which SR-Dec can answer YES, in order for different proofs on the hardness of SR to be provided based on other existing results, such as the one by (Guruswami & Vardy, 2005), as suggested by reviewer **KHax**
>
>
> ### Requested changes
> > A bit nitpicking on the wording: "this is already problematic for any SR algorithms". I wouldn't say that this is problematic because SR algorithms are nevertheless fairly successful. Maybe let's say that it already poses a challenge to do this on the training set efficiently.”
>
> Thank you, we agree and improved the formulation as suggested.
>
> >Is it guaranteed that there is a unique solution to Equation 1? If not, I wonder whether it should be f^* \in \argmin since the task would not to find all solutions, but only one of them.
>
> The reviewer is right that multiple optima might exist. Still, to the best of our knowledge, it remains commonplace to use $f^* = $ argmin rather than $f^* \in $ argmin.

---

> > ### Comment · Reviewer_YHbt · 2022-09-07
> > **Argmin**
> >
> > Thanks for the improvements.
> >
> > >  it remains commonplace to use $f^* = argmin$ rather than $f* \in argmin$
> >
> > I agree that the former is more common, but unfortunately, that doesn't mean that it is correct.
> > For simplicity, please check Wikipedia. Argmax (and thus also argmin) is defined to return a set of all optima and not a single arbitrary element of this set.
> > https://en.wikipedia.org/wiki/Arg_max

---

> > > ### Author Response · Authors · 2022-09-08
> > > **Response to Reviewer YHbt on Argmin**
> > >
> > > Thank you, the reviewer is right. We will change to $\in$ as suggested.

---

### Review · Reviewer_tATo · 2022-08-10

**Summary Of Contributions:**

The authors prove that symbolic regression (SR) is NP-hard by reduction to the unbounded subset sum problem.  The reduction holds also for cases in which constants (and distributions) appear in the search space.

**Broader Impact Concerns:**

This paper poses no ethical concerns.

**Requested Changes:**

- Highlight the significance of your result by briefly discussing what consequences you expect it to have / what benefits you expect it to bring to research on SR.

- Briefly discuss how the two theorems depend on the choice of search space.  The reduction seems to show that allowing for just sum, product, and equality is enough to trigger NP-hardness, but it doesn't say much for SR cases in which these operations are not allowed.  Please briefly clarify this point.

- Briefly discuss links to classical hardness results in ML.

- p 2: "Early forms of GP where proposed" -> were.

- p 3: Search space of SR: The definition depends on the notion of composition between functions, which is defined, and on that of composition between functions and variables, which is not.  Please clarify.

- p 4: "i.e., distance" -> a distance

**Strengths And Weaknesses:**

+ Well written, accessible even to non-expert readers.  The proofs are easy to follow.

+ Good coverage of the SR literature.

+ The mathematical setup and the proofs themselves seem sound to me.

+ The contribution seems to be novel.

- Missing discussion of significance and expected impact of this result.  The authors mention that the paper settles a well-known conjecture, which is true.  In a sense, the paper serves to retroactively justify the usage of genetic algorithm and other heuristic search procedures for this problem.  However, the significance of doing so is a bit unclear.  A more detailed discussion of the limits of this result and of its dependency on the choice of search space would have helped to give this result more depth.

- Lacks connection to classical hardness results in statistical learning.  Specifically, considering that SR is cast essentially as an empirical risk minimization problem, I was expecting the authors to explore the connection between their results and the well-known hardness of ERM.  Could ERM be reduced to SR?

---

> ### Author Response · Authors · 2022-09-06
> **Response to Reviewer tATo**
>
> We thank the reviewer for assessing our work. We find that the requested changes nicely map to the identified weaknesses, thus we proceed by answering the requested changes alone.
>
> ### Requested changes
> > Highlight the significance of your result by briefly discussing what consequences you expect it to have / what benefits you expect it to bring to research on SR
>
> We have extended our conclusion to incorporate the requested change. We now explain that our work settles the question as to whether SR is NP-hard and, this way, we hope to kick-start the study of whether interesting bounds can be achieved for specific instances / variants of SR. In fact, while we started from a general definition and performed restrictions that allowed us to prove hardness, there might exist instances of SR (e.g., choices of P, loss functions, types of data) that are particularly interesting for practical uses of SR and for which better runtimes might be possible.
> In our view, future research efforts in this direction would nicely complement the historical way of performing research in the field of SR, which has been focused on heuristic algorithmic development.
>
> > Briefly discuss how the two theorems depend on the choice of search space. The reduction seems to show that allowing for just sum, product, and equality is enough to trigger NP-hardness, but it doesn't say much for SR cases in which these operations are not allowed. Please briefly clarify this point.
>
> The reviewer is right that our proofs depend on choosing a specific set P of functions and thus a specific induced search space. To prove hardness, it suffices to find some instances of the problem to which all other instances of an already-known-to-be-hard problem can be mapped.
> Indeed, other instances of SR might exist, e.g. with other search spaces or losses, that can be solved in polynomial time.
> Our intention was to settle the matter on the fact that SR, in its most general case, is hard (since it contains hard instances).
>
> We tackled this comment as follows:
> 1. We acknowledge this aspect at the end of the background section, explaining that we are able to prove hardness by imposing the functions that SR can explore to be inherently discrete in nature (which triggers hardness even under losses that are not the 0-1 loss, the latter being common in ERM literature on hardness).
> 2. We elaborate on the scope of our contribution in the conclusion, as answered in the previous question. Here we made choices (including P and thus the search space) useful to settle the matter of whether SR is in fact NP-hard.
> 3. In the new version, we also included a remark at the end of Section 4 sketching how, thanks to a regularization function $C$, one can further change the search space for which SR-Dec admits a YES answer.
>
>
>  > Briefly discuss links to classical hardness results in ML
>
> Thank you for asking us to do this, we believe this helped nicely set our paper in a wider context. We have split section 2 Background into two subsections, with section 2.2 featuring the requested discussion. There, we refer to related results in empirical risk minimization (ERM), from the hardness of training 3 node neural networks (Blum & Rivest, 1992) to polynomial bounds for kernel-based and neural-based methods (Feldman et al. 2012), and to results in coding theory (Guruswami & Vardi, 2005) as suggested by reviewer KHax. Hopefully, this meets the reviewer’s satisfaction.
>
> We have also re-formulated the paper to be in line with ERM terminology.
>
> > p 2: "Early forms of GP where proposed" -> were.
>
> Fixed, thank you.
>
> > p 3: Search space of SR: The definition depends on the notion of composition between functions, which is defined, and on that of composition between functions and variables, which is not. Please clarify.
>
> We clarified these notions. Variables are the arguments of functions and to the best of our knowledge, one needs not to define a composition between the two. Yet, thanks to identity functions one can consider variables as functions themselves. This allows us to talk about composition between functions to generally describe SR. We have clarified this at the beginning of Section 3.
>
> > p 4: "i.e., distance" -> a distance
>
> The reviewer is right, thank you. We have reformulated what we mean by loss (to operate on one observation rather than on the training set) to better frame our work in terms of ERM. Consequently, this piece of text no longer appears.

---

### Review · Reviewer_KHax · 2022-08-27

**Summary Of Contributions:**

The paper studies the complexity of the problem of symbolic regression from data to learn a mathematical expression and shows that this problem is NP hard. While it has been widely believed that this problem is NP-hard (Lu et al 2016, Peterson et al 2019), the paper claims to be the first to provide a formal proof for the NP-hardness.



**Broader Impact Concerns:**

There are no broader impact concern for the paper. The authors are advised to add a broader impact statement building on their argument about the benefits of symbolic model in the introduction. But even without this statement, the broader impact of studying the complexity of symbolic regression is obvious.

**Requested Changes:**

Definition 2 can be better worded: "formed by compositions of elements of P and their compositions" - composition is naturally treated as being recursive and unless, one wants to restrict composition to just two (or fixed number of) levels - there is no need to mention compositions twice. Page 4 anyway talks about restrictions on the space of f (in particular, rules out recursion) later.

In Definition 3, is P including the ephemeral random constant / distribution representing element R or is it just the set of functions and variables. If it is just the later (as the wording of Definition 3 suggests), the discussion of R before Definition 3 appears to be a distraction that can be eliminated from discussion.






**Strengths And Weaknesses:**

Strengths

1. The paper is well-written. The discussion of related work is comprehensive. A strong case is built for the need to learn interpretable symbolic models that can be analyzed for safety. The paper also identifies papers where the NP-hardness of symbolic regression has been hinted or mentioned before. A detailed discussion on symbolic regression, SRbench and deep learning based SR is presented.

Weaknesses

1. The reviewer was a bit surprised that no prior work has proven symbolic regression to be NP-hard. It appears at the end of Page 3 and start of Page 4 that the SR problem definition (which would ideally differentiate between training on a set of data and generalizing to a validation set) is replaced with just the minimization of the loss function. At this point, prior work such as https://arxiv.org/abs/cs/0405005 become an instance of (1) and the result of say, "Maximum-likelihood decoding of Reed-Solomon Codes is NP-hard" ("computing the degree k polynomial that disagrees with the minimum number of input point") would imply the main result in the paper by suitably defining the L function. So, the reviewer is not convinced the main result in the paper is new.

2. The reduction from unbounded subset sum problem to SR is interesting. But whether epsilon is part of the problem definition of SR or part of the solution is debatable. One can argue that the goal is to find f that minimizes epsilon, but the problem definition itself does not set the value of epsilon (something that is true in practice). Under this argument, it would not be okay to pick epsilon to be 0 when translating USSP-Dec to SR.

3. See some specific requests on presentation in "Requested Changes".

---

> ### Author Response · Authors · 2022-09-06
> **Response to Reviewer KHax**
>
> We thank the reviewer for the valuable comments. We will firstly answer to the points identified as weaknesses, and then to those identified as requested changes.
>
> > The reviewer was a bit surprised that no prior work has proven symbolic regression to be NP-hard. It appears at the end of Page 3 and start of Page 4 that the SR problem definition (which would ideally differentiate between training on a set of data and generalizing to a validation set) is replaced with just the minimization of the loss function. At this point, prior work such as https://arxiv.org/abs/cs/0405005 become an instance of (1) and the result of say, "Maximum-likelihood decoding of Reed-Solomon Codes is NP-hard" ("computing the degree k polynomial that disagrees with the minimum number of input point") would imply the main result in the paper by suitably defining the L function. So, the reviewer is not convinced the main result in the paper is new.
>
> Sincerely, we were surprised too that SR had not proven to be NP-hard yet. We believe this is most likely due to the fact that the field has been relatively small and, in our view, it has gained more interest only recently, due to a renowned interest for interpretable ML.
> We understand that the reviewer raises two points here: (1) absence of considerations for generalization in the definition of the SR problem, and (2) potential lack of novelty. We respond in order.
>
> 1. About minimizing the loss on the training set alone, we made this choice because it is commonly made in the literature concerning proving NP-hardness. For example, this is the case for the related works we included in the new section 2.2 (requested by reviewer **tATo**), on the NP-hardness of empirical risk minimization (ERM). There, we included the suggested piece of work by Guruswamy & Vardi, thank you. Overall, we would defend the choice to focus on ERM because, although one wishes to have a generalizing model, firstly one needs to optimize the model to the training set to a sufficient level (i.e., past under-fitting) before being able to start overfitting.
>
> We have acknowledged that validation data is used in practice, adding the following line after Def. 3:
>
> *“A common practice is to heuristically use a separate set of data (the validation set) to estimate the generalization to observations
> outside the training set”*
>
> 2.  The reviewer is absolutely right that we may frame certain instances of SR to operate similarly to the MLD-RS problem defined by Guruswamy & Vardi. As such, we could have proven that SR is NP-hard because MLD-RS is. We thank the reviewer and have sketched how such sort of proof can be achieved at the end of section 4. The reviewer states that this may have a negative impact on the novelty of our result. To strengthen our result, we now explain in Sec. 2.2. how we need not use a 0-1 loss to prove that SR is NP-hard. We explain that we can achieve a “0-1 loss-like effect” due to restricting SR to seek functions that are inherently discrete by opportunely choosing $\mathcal{P}$. This said we would still like to remark that:
> - NP-hardness proofs may be achieved with (restrictions &) reductions from different problems. We are not sure that the fact that a different problem to reduce from exists makes our result less interesting.
> - TMLR’s guidelines state: *‘[…] Nor should it form the basis for rejecting work on a method considered not “novel enough”, as novelty of the studied method is not a necessary criteria for acceptance.’* As SR was believed to be NP-hard for a long time, we agree that it is not surprising that this was the case. Quoting TMLR’s guidelines again: *‘We […] focus instead on the notion of “interest”’*. We hope that the reviewer finds our effort to provide a first proof (even though different sorts of proofs can indeed be given, we totally agree) to be of sufficient interest.
>
> > The reduction from unbounded subset sum problem to SR is interesting. But whether epsilon is part of the problem definition of SR or part of the solution is debatable. One can argue that the goal is to find f that minimizes epsilon, but the problem definition itself does not set the value of epsilon (something that is true in practice). Under this argument, it would not be okay to pick epsilon to be 0 when translating USSP-Dec to SR.
>
> We would like to answer this in two parts: (1) having epsilon at all in the definition of SR-Dec; and (2) picking epsilon = 0 specifically. We do this in the following comment due to a lack of space here.

---

> > ### Author Response · Authors · 2022-09-06
> > **Response to Reviewer KHax (part 2)**
> >
> > 1. The reviewer is right: the goal is to minimize epsilon. That is implicit in the definition of the minimization version of the problem, which we simply refer to as the *SR problem*. For the decision version, *SR-Dec*, we make use of epsilon to transform the minimization problem into a decision problem. It seems to us that this is standard. For example, in Kellerer et al. 2004 *"Knapsack problems"*, decision versions of minimization (or, equivalently, maximization problems), commonly impose inequalities or equalities w.r.t. a certain threshold.
> > To cite the classic example of Knapsack, arguably the user wishes to have a minimal weight knapsack with maximal value; however, the decision version asks that the weight is below a specific threshold. Similarly, for the case of training a neural network, Blum & Rivest 1992 (cited in the new Section 2.2) show NP-completeness of the decision version of whether the network can correctly label *all* the observations, i.e., they impose an error of 0. The MLD-RS problem referenced by the reviewer is also a decision problem, that uses an inequality w.r.t. a variable $w$, then again the ultimate aim is to minimize $w$ (from the NP-completeness of the decision version of the problem follows the hardness of the minimization version of the problem).
> >
> > 2. We are not sure why picking $\epsilon = 0$ might not be OK in principle. To prove that SR-Dec is NP-complete (and, consequently that the minimization version is NP-hard), we need to show that there exist at least certain instances of setting the parameters of the problem such that reduction is possible from another problem that is already known to be NP-complete.
> > This said, we acknowledge that picking $\epsilon = 0$ essentially makes the loss, even if it is, e.g., the squared error loss, essentially become a 0-1 loss. Thus, we strived to improve the interestingness of our result (although we would defend that this is not formally necessary) to show that NP-hardness can be achieved for other values than $\epsilon = 0$. As mentioned before, we are able to do this under more general losses than the 0-1 loss because we can restrict SR to operate on functions that are inherently discrete in nature.
> > - We now show that Theorem 1 can be proven for instances of SR-Dec using arbitrary $0 \leq \epsilon \leq 1$.
> > - We note that our proof for Corollary 1 needed to change and it is a little bit more involved now. We are still able to show that there exist instances (i.e., values of $\epsilon$), such that even when a real-valued constant with arbitrary value *must* be used, the constant needs necessarily be set to 0 for SR-Dec to answer YES. This allows us to reduce from USSP-Dec as shown in the proof of Theorem 1.
> >
> > ### Requested changes
> >
> > > Definition 2 can be better worded: "formed by compositions of elements of P and their compositions" - composition is naturally treated as being recursive and unless, one wants to restrict composition to just two (or fixed number of) levels - there is no need to mention compositions twice. Page 4 anyway talks about restrictions on the space of f (in particular, rules out recursion) later.
> >
> > We thank the reviewer and did as suggested.
> >
> > > In Definition 3, is P including the ephemeral random constant / distribution representing element R or is it just the set of functions and variables. If it is just the later (as the wording of Definition 3 suggests), the discussion of R before Definition 3 appears to be a distraction that can be eliminated from discussion.
> >
> > Thank you for pointing this out. We acknowledge that this point was unclear because we wrote that $\mathcal{P}$ contained functions and variables, while not mentioning constants. We did mean that $\mathcal{P}$ can contain constants because they can be represented with constant functions. To address also a similar comment by reviewer **tATo** regarding variables, we now clarify at the beginning of section 3 that both variables and constants can be considered to be particular functions, and proceeded with saying that $\mathcal{P}$ contains functions.
> >
> > ### Broader impact concerns
> > > There are no broader impact concern for the paper. The authors are advised to add a broader impact statement building on their argument about the benefits of symbolic model in the introduction. But even without this statement, the broader impact of studying the complexity of symbolic regression is obvious
> >
> > We preferred not to include the section on broader impact because the guidelines of TMLR ask to include it if the work carries a significant risk of harm. We do, however, agree that we should better discuss the meaning of our work. We the conclusion to that end. We now better explain that our work intends to settle the point that SR is in fact NP-hard. We add that, by doing so, we hope to initiate the discussion on whether there may exist specific settings of SR that are interesting in practice and for which better bounds can be derived.

---

> > ### Comment · Reviewer_KHax · 2022-09-13
> > **Thank you**
> >
> > "The reviewer is absolutely right that we may frame certain instances of SR to operate similarly to the MLD-RS problem defined by Guruswamy & Vardi. As such, we could have proven that SR is NP-hard because MLD-RS is. We thank the reviewer and have sketched how such sort of proof can be achieved at the end of section 4. The reviewer states that this may have a negative impact on the novelty of our result. To strengthen our result, we now explain in Sec. 2.2. how we need not use a 0-1 loss to prove that SR is NP-hard. "
> >
> > The reviewer appreciates the addition of Section 2.2 and inclusion of the discussion in the review.
> >
> > The argument in going from 0-1 loss to continuous loss is not clear. Note that IP is NP-complete and LP is in P because the IP problem instances come with an additional requirement that the solutions must be integers. LP does not. Similarly, there is no reason to force SR solutions to be discretized.

---

> > > ### Author Response · Authors · 2022-09-14
> > > **Response to KHax (additional comment)**
> > >
> > > Thanks to the reviewer for their comment. Indeed, we have updated the last part of Sect 2.2 to avoid the confusion that might arise from the use of term `discreteness' -- this term has now been removed (see rev2). In this paper we have shown that a general SR formulation is hard but surely we acknowledge that there might be certain choices of the set of primitives for which SR is in P. We hope that this paper may spark more research on upper and lower bounds of instances with such certain choices of the set of primitives; or generally of other SR variants of practical interest. This discussion has been added in the last part of the Conclusion during the previous revision (see rev1).

---

### Decision · Action_Editors · 2022-10-11

**Recommendation:** Accept as is

**Comment:**

The technical tools used in the proof do not seem especially complex
or novel, and the conclusion is also not surprising. That said, the
reviewers and I feel that a formal proof of a key fact about symbolic
regression is of inherent value. The paper is also well-written, and
the discussion of related work is quite substantive. The authors also
did a good job of revising the paper based on the reviewers' comments.
Given all this, I am recommending acceptance as is.

**Audience:**

Symbolic regression is a classic problem with a broad range of
applications in science and engineering. This foundational result
should interest a nontrivial subset of the TMLR audience.

**Claims And Evidence:**

The paper offers a proof of the NP-hardness of symbolic
regression. The problem has been believed to be NP-hard for some time,
but this appears to be the first time this claim has been formally
proven. The proof is based on a reduction from the Unbounded Subset
Sum problem and is correct (though relatively simple).